# Posture Correction Therapy and Pelvic Floor Muscle Function Assessed by sEMG with Intravaginal Electrode and Manometry in Female with Urinary Incontinence

**DOI:** 10.3390/ijerph20010369

**Published:** 2022-12-26

**Authors:** Katarzyna Jórasz, Aleksandra Truszczyńska-Baszak, Aneta Dąbek

**Affiliations:** Physiotherapy Department, Józef Piłsudski University of Physical Education in Warsaw, 00-968 Warsaw, Poland

**Keywords:** stress urinary incontinence, body posture, pelvic floor muscles, corrective therapy

## Abstract

Introduction: The aim of the study was to assess the influence of the implemented therapeutic programme, which consisted of body posture correction and of change of habits, on the pelvic floor muscle function in women with stress urinary incontinence. Material and methods: The 60 women were randomly divided into two groups: the study population and the clinical control group (subjects received envelopes with numbers of the group: 1- study population aged 38.3 ± 5.54, 2- clinical control group aged 35.5 ± 4.7. We used the following research methods: A personal questionnaire with subjects’ demographics and with questions related to the type of work, physical activity, childbirths and any issues related to the pelvic floor (pre-test), Pelvic floor muscle assessment with the use of the PERFECT Scheme and the Oxford scale palpation examination, sEMG with intravaginal electrode and manometry with an intravaginal probe—pre-test and post-test. Subjective assessment of body posture in the sagittal plane according to the McKenzie methodology. Results: In both groups, VRP (resting vaginal pressure) and resting PFM tension were significantly reduced. The strength and endurance of PFM, tension during MVC and VSP (intravaginal pressure during contraction) increased, with no difference between the groups. SUI decreased significantly, and quality of life improved significantly in both groups. Conclusions: Education of the pelvic floor and changing habits significantly affected the activity of PFM and improved the quality of life in the group of patients with SUI. The posture correction therapy with manual therapy and stretching exercises did not increase this effect.

## 1. Introduction

Proper body posture in everyday activities is maintained by the muscles of the so-called muscle cylinder. Pelvic floor muscles are some of the muscles that create this cylinder [1]. The structures of the muscle cylinder tense up for 35–45 ms before any move is made (the *feedforward* mechanism). The feedforward tension of the proprioceptive muscles serves to stabilise the spine and the pelvis. It also allows for containing urine and faeces when pressure within the abdominal cavity rises, e.g., during coughing. The set-up of individual body segments impacts the function of the cylinder structures, including pelvic floor muscles [2,3,4].

The impact of body posture on pelvic floor muscles has been the subject of numerous studies [5,6,7,8,9,10,11,12,13,14].

A review of the literature revealed a lack of reports on the influence of posture correction therapy on pelvic floor muscle function.

The aim of the study was to assess the influence of the implemented therapeutic programme, which consisted of body posture correction and of change of habits, on the pelvic floor muscle function in women with stress urinary incontinence.

## 2. Material and Method

The consent to conduct the study was issued by the Senate Research Ethics Committee of the University of Physical Education in Warsaw on 23 November 2018 (no SKE 01-32/2018). The study protocol was also registered in the US National Library of Medicine of the National Institutes of Health, no NCT04366557. The tests were conducted at the Profemed Bobrowiecka outpatient clinic in Warsaw between 2 January 2019 and 31 March 2020. The study involved 60 women with stress urinary incontinence.

The qualified subjects were randomly divided into two groups: the study population and the clinical control group (subjects received envelopes with numbers of the group: 1- study population, 2- clinical control group).

The inclusion criteria were: age between 25 and 45 years, female sex, diagnosed stress urinary incontinence and written consent to participate in the study.

The exclusion criteria were: instrumental delivery, unplanned caesarean section, large birth weight of the baby (over 4 kg), prior surgery to the abdomen (other than a caesarean section and appendectomy and laparoscopy), prior gynaecological surgery, chronic cardiovascular and pulmonary diseases, frequent infections of the lower urinary tract, prior neurological incidents (e.g., stroke), injuries to the pelvis or the spine, overweight, obesity and menopause.

Statistical analysis (Mann-Whitney U test) did not reveal significant differences between the groups in regard to age, body mass, body height or BMI. Statistical significance was set at *p* ≤ 0.05. There were differences between the groups in regard to age (*p* = 0.047), yet within the studied age range, it did not affect results.

Table 1 presents descriptive statistics (Table 1).

Women from the study population had a mean number of childbirths of 1.8 ± 0.7 (vaginal birth 1.5 ± 0.9; CC 0.33 ± 0.66), and in the clinical control group, it was 1.8 ± 0.9 (vaginal birth 1.6 ± 0.9; CC 0.27 ± 0.45). There was no significant difference between the groups in regard to the number of given births. Among the subjects, 52 (86.67%) had sedentary work and lived in a city. With regard to physical activity, 39 women (65%) assessed their physical activity as low (no physical activity or exercising on their own maximum twice a week), 15 women (25%) assessed their physical activity as average (three times a week for a minimum of 45 min), 6 (10%)—as high (training more than 3 times a week for a minimum of 45 min).

Both groups were assessed twice—before and after the implemented therapy (in the study population).

We used the following research methods:(1)A personal questionnaire with subjects’ demographics and with questions related to the type of work, physical activity, childbirths, and any issues related to the pelvic floor (pre-test).(2)Pelvic floor muscle assessment with the use of the PERFECT Scheme and the Oxford scale palpation examination, sEMG with an intravaginal electrode (MyoPlus Pro electromyograph produced by Neurotrack, Verity, Ireland) and manometry with intravaginal probe (Myo200 manometer produced by Gymna, Bilzen, Belgium)—pre-test and post-test. For the EMG test of each of the subjects, a separate intravaginal probe Periform Plus was used (produced by Neen Performance Health, Nottinghamshire, UK).(3)Subjective assessment of body posture in the sagittal plane according to the McKenzie methodology—mechanical diagnosis and therapy (ear lobe projection)—used in the therapeutic part (information for the person performing the therapy as well as education for the subject on how to assume the correct body posture).

The assessment of the muscles of the perineum began with palpation according to the PERFECT Scheme (Table 2). When conducting the examination, attention was paid to the correctly performed contraction of pelvic floor muscles. The palpation was followed by a ten-minute break allowing muscles to relax.

The next stage of pelvic floor muscle assessment was EMG with an intravaginal electrode. The equipment used for the EMG test was a single-channel MyoPlus Pro (Neutrotrack, Verity, Ireland) with software for registration and analysis of the results.

The intravaginal electrode was disinfected prior to use with a dedicated non-alcoholic disinfectant. Ultrasound water-based gel was used for the application. The electrode was inserted into the vagina to the point marked on the probe. The reference electrode was placed in the area of the right anterior superior iliac spine. Before the measurement, there was a minute wait for the signal to stabilise, and then the subject was asked for an initial contraction and relaxation of the muscles. This served to check whether the equipment was working properly. Furthermore, it was checked that the subject did not push the electrode out during the contraction and that she performed the contraction correctly (without holding her breath). Then the subject was asked to perform a series of five five-second-long contractions with ten-second long pauses between them (in line with the equipment software scheme). There was a sound signal for each contraction and each relaxation. The parameters analysed in the study were the mean tension during the maximal contraction (any of the performed contractions) and the mean tension during relaxation. The software averaged the results.

After another 10 min break, the manometry test was performed. It used a silicone pressure probe (Laborie, 11 cm) and registration equipment of Myo200 (Gymna, Bilzen, Belgium). A non-latex cover was placed on the probe before the probe was inserted into the vagina. Water-based gel was used for the application. Before application, the equipment was calibrated to zero. The unit of measurement for the manometer test was millimetres of mercury (mmHg). The parameters analysed in the study were: intravaginal pressure at rest (registered after a minute since the placement and adjustment of the probe) and the highest intravaginal pressure during contraction (one of three trials).

After the diagnostic tests, the subjects from the clinical control group followed general educational recommendations for a period of 6 weeks. Subjects from the study population had additional 6-week posture correction therapy (details are presented further in the paper). An activity diary was used to monitor the recommended exercise. Study population subjects marked exercises which they performed on a given day with an “x”; it provided them with additional motivation for regular exercise. After 6 weeks, the diagnostics of pelvic floor muscles were repeated.

### 2.1. Education

Both groups were involved in the education part of the study. Education concerned with pelvic floor and correct habits. It went according to the following scheme:(1)Information on the anatomy and function of pelvic floor muscles. Anatomy chart presentation.(2)Perception exercises: self-palpation of bony anatomical landmarks (anterior superior iliac spines, the ilium, the sacrum, the coccyx, the ischial tuberosities, the pubic symphysis) when seated in a chair, with or without a tennis ball.(3)The activity of pelvic floor muscles during activities of daily life: sitting and standing positions and their corrections, positions to take when sneezing/coughing, in the toilet, and during physical activity.(4)Learning the correct activation of pelvic floor muscles in lower positions and without load (depending on the needs of the assessed subject); instructions “exhale and then tighten the anus and the urethra, then pull your perineum up, as if you wanted to pull in a tampon into your vagina”.(5)Learning how to correct one’s posture during activities of daily living.

### 2.2. Posture Correction Therapy

The study population were involved in posture correction therapy. The therapy lasted 6 weeks and consisted of individual manual therapy and exercise at home.

Manual therapy involved chosen soft tissues. Therapeutic sessions lasted for 45 min and took place once a week. The physiotherapist focused on muscles which affect posture in the sagittal plane and, according to the idea of anatomy trains, affect the pelvic floor muscles [15]. These muscles are subscapularis muscles, sternocleidomastoid muscles, pectoralis major, diaphragm, iliopsoas muscles, quadratus lumborum muscles, and biceps femoris muscles. Selected methods of soft tissue therapy were used: myofascial release, trigger points therapy and deep tissue massage. The use of a particular technique depended on the sensitivity of the individual’s tissue and was carried out according to the scheme described in the literature (Figure 1).

Each time, a postural assessment in the sagittal plane was conducted prior to therapy. When assessing posture, special attention was paid that the gymnastic stick marked a straight line from the lateral malleolus through the middle of the knee, trochanter major, middle of the thoracic kyphosis, acromion, and ear lobe (Figure 2).

Exercise recommended for the home involved stretching of the muscles that were treated in the manual therapy—examples of exercise are presented in illustrations (Figure 2 and Figure 3). An additional type of exercise was a corrective activity, according to the methodology by McKenzie, performed in a standing position: Take a hunched position for two seconds. Then correct your posture, maximally straightening your body. Maintain the physiological curvatures of your spine. You may feel the tension in your lower back and/or within your shoulders. Ease this hypercorrection by 10–15%. Repeat this exercise 10 times, several times a day. The subjects were instructed to exercise every day and to register this activity in the training diary.

Flow-chart (Figure 4) presents stages of the assessment, division into groups and methods used.

### 2.3. Statistical Analysis

To verify the study hypotheses, we conducted a statistical analysis. We used the IBM SPSS Statistics software package version 26 to analyse the essential descriptive statistics and to conduct the test for normal distribution and a series of multi-factor variance analyses in mixed schemes. The level of statistical significance was set at α < 0.05.

## 3. Results

We analysed the essential descriptive statistics and conducted the Shapiro-Wilk test for normality. The result of this test was statistically significant for the majority of the analysed variables. This meant that the distribution was divergent from the Gauss curve. However, the value of asymmetry did not exceed the conventional absolute value of 2 in any of the cases [16]. On this basis, the distribution of the analysed variables may be regarded to be close enough to normal distribution to enable parametric statistical tests. The table presents the values of all the calculated statistics along with the normality test for control group (Table 3) and for study group (Table 4).

### 3.1. The Implemented Therapeutic Programme and Pelvic Floor Muscles at Rest

We conducted variance analysis in the mixed scheme. In the analysed scheme, the intergroup variable belonged to the study population or to the clinical control group, and the within-group variables were measurements taken prior to and after the implemented therapy. The result revealed a statistically significant within-group effect only. This means that the subjects had intravaginal pressure at rest significantly higher before therapy than after therapy. This effect explained 46% dependent variable variance. Figure 5 illustrates the contrasted means (Figure 5).

Then we conducted an analogical analysis of variance, which contrasted tension at rest, measured by surface electromyography (sEMG). The result revealed a statistically significant within-group effect only. Subjects had tension at rest significantly higher before therapy than after therapy. This effect explained the 30% dependent variable variance. Figure 6 illustrates the contrasted means (Figure 6).

### 3.2. The Implemented Therapeutic Programme and Pelvic Floor Muscle Tension Parameters

To assess the effect of the implemented therapy on chosen parameters of pelvic floor muscles in the MVC time (the second hypothesis), we conducted a series of multi-factor variance analyses in mixed schemes. First, we compared the power parameter (P) assessed within the PERFECT Scheme palpation with the Oxford scale (0–5). The result revealed a statistically significant effect of the implemented therapy only. This means that the MVC power was significantly lower before therapy than after the implemented therapy. This effect explained 56% dependent variable variance. Figure 7 illustrates the contrasted means (Figure 7).

We conducted an analogical comparison of endurance of pelvic floor muscle contraction (E in the PERFECT Scheme), measured in seconds of maintaining the contraction (1–10). The results revealed both the statistically significant effect of the implemented therapy and the statistically significant effect of interaction. This means that the pelvic floor muscle contraction endurance was significantly higher after therapy than before therapy (this effect explained 64% dependent variable variance). The effect of interaction meant that before therapy, there were no differences between the groups. It explained the 10% variance. After intervention, the clinical control group had statistically better PFM endurance than the study population. Figure 8 illustrates the contrasted means (Figure 8).

We compared intravaginal pressure during contraction. Results revealed a statistically significant effect of implemented therapy and of interaction. Intravaginal pressure during contraction was significantly higher after therapy than before therapy (56% variance). In the study population, the difference resulting from implemented therapy was greater according to the effect of interaction (20% dependent variable variance). Figure 9 illustrates the contrasted means (Figure 9).

We conducted an analogical comparison, this time for mean tension during contraction (sEMG). Results revealed a statistically significant effect of implemented therapy and the effect of interaction. Subjects after therapy had significantly higher mean tension during contraction. The effect of therapy explained a 22% variance of tension during contraction. Study population subjects had significantly higher mean tension after the implemented therapy, while clinical controls did not differ in subsequent measurements. The effect of interaction explained 7% dependent variable variance. Figure 10 illustrates the contrasted means (Figure 10).

## 4. Discussion

Posture correction aimed to equalise the myofascial tension with manual therapy and stretching exercise. We used the definition of correct posture as posture, which ensures the balance of antagonistic muscle groups [17,18,19]. The position of individual body segments was assessed with physiotherapeutic examination with McKenzie methodology—mechanical diagnosis and therapy—ear lobe projection [20]. The intention was to reflect a therapist’s everyday work with patients. The structures that were subject to therapy were chosen on the basis of a literature review [15].

The effect of postural therapy on pelvic floor muscles was assessed in a few studies only. Kasper-Jędrzejewska et al. [21] assessed the effect of Structural Integration (Rolfing). They noted a positive effect of therapy (10 sessions according to their methodology) on the bioelectric activity of muscles of the perineum at rest. Pelvic floor muscle activity parameters during contraction improved, too.

In their study, Sheikhhoseini and Massoud Arab [22] proved the impact of dry needling of myofascial tapes on perineum muscle relaxation. Their study population were men suffering from excessive tension of pelvic floor muscles. After therapy, the increased tension at rest decreased. This positively impacted muscle function and resulted in the easing of symptoms related to problems with urination.

Similarly, MartinsReis et al. [23], based on their latest studies, noted the impact of body posture on muscles and stressed the necessity of posture correction in the course of therapy. They assessed body posture in women with urinary incontinence. In addition, one of the studied groups consisted of women with diagnosed myofascial dysfunction within pelvic floor muscles (pain in palpation). The authors used photometry to assess posture. They found that women with urinary incontinence and pelvic floor muscle dysfunction had a greater forward inclination of the pelvis.

The impact of body posture on the bioelectric activity of pelvic floor muscles was described in studies in which exercise was based on whole-body activation and correction of body posture, e.g., Pilates. Chmielewska et al. [24] presented the effectiveness of such exercise. They compared their results with the results of a group of women who used biofeedback in their pelvic floor muscle training. Pilates training had much better results.

### 4.1. Electromyography

Electromyographic assessment of pelvic floor muscles in our study showed much higher results of muscle tension at rest than average, and there was no statistically significant difference between the two groups. After implemented therapy, this tension lowered significantly, and again there was no significant difference between the study population and controls. This means that education on correct habits influenced the initial tension in pelvic floor muscles. The reason behind the lasting increased tension of pelvic floor muscles at rest may have been the limited time between assessment (before and after) in the conducted measurements (6 weeks). In the future, it may be worth measuring the effect after, e.g., three or six months.

Our results are similar to the results of other authors. In the literature review, we found studies which had noted higher than normal mean tension of perineum muscles among studied women. This is a common issue, especially among active young females. The reasons quoted for hypertonic PFM are, inter alia, stress, high physical activity, and incorrect body posture [23,25].

In our study, the group who had posture correction therapy had statistically higher muscle tension in maximum voluntary contraction. Kasper-Jędrzejewska et al. [21] had similar results—in her study, after Structural Integration (Rolfing), the mean tension of pelvic floor muscles increased.

### 4.2. Palpation Examination

There are several types of palpation assessment of PFM. The PERFECT Scheme is the most popular type of assessment [26,27].

The author (KJ) has the appropriate qualification and clinical practice to conduct the assessments. This is important in the context of applying this method. To avoid being biased by earlier assessments (from the first test), prior to the second assessment, the therapist (KJ) purposefully did not check the initial assessment results. Two elements of the assessment were included in the statistical analysis: power (Oxford Scale 0–5) and endurance (seconds of maintaining contraction 0–10).

After six weeks, both groups had a significant increase in the power of PFM, and there were no significant differences between groups in this respect (and increase of endurance, with significant difference between groups). Interestingly, there was greater improvement in the clinical control group. We did not find any studies to which we could relate these findings.

### 4.3. Manometry

The measurements revealed a significant decrease in intravaginal pressure at rest in the second test in both groups, without any significant differences between the groups. There was also a difference in the intravaginal pressure during the maximum voluntary contraction, which significantly increased in both groups, significantly more in the study population.

Value of the study. No similar studies are available in the literature, so we are unable to refer to any similar results. Nobody before us conducted a manometric assessment of this group of muscles before and after posture correction therapy. However, our results are in line with the impact of pelvic floor muscle training on the manometry results [28].

The therapy implemented in the study can be conducted by any professional therapist, regardless of the training courses they have completed. Paying attention to body posture and education opens new opportunities for the treatment of inter alia stress urinary incontinence. It gives a chance to make therapy more effective.

Study limitation. For future studies, it seems worth arranging pelvic floor muscle assessment at the same time of the day. PFM may function differently in the afternoon hours (overload after the whole day) than in the morning hours. It may also be worth enlarging the studied population, organising a multi-centre study and prolonging therapy to a minimum of 12 weeks.

## 5. Conclusions


(1)Education of the pelvic floor and change of habits statistically significantly affect the activity of PFM. It decreases PFM tension at rest and increases PFM power and endurance during MVC.(2)Posture correction therapy in the sagittal plane (manual therapy and stretching exercise) did not have a significant impact on PFM in comparison to education in the study population. We recommend in the therapy of patients with stress urinary incontinence of the I and II degree to start physiotherapy with education and training in autotherapy.


## Figures and Tables

**Figure 1 ijerph-20-00369-f001:**
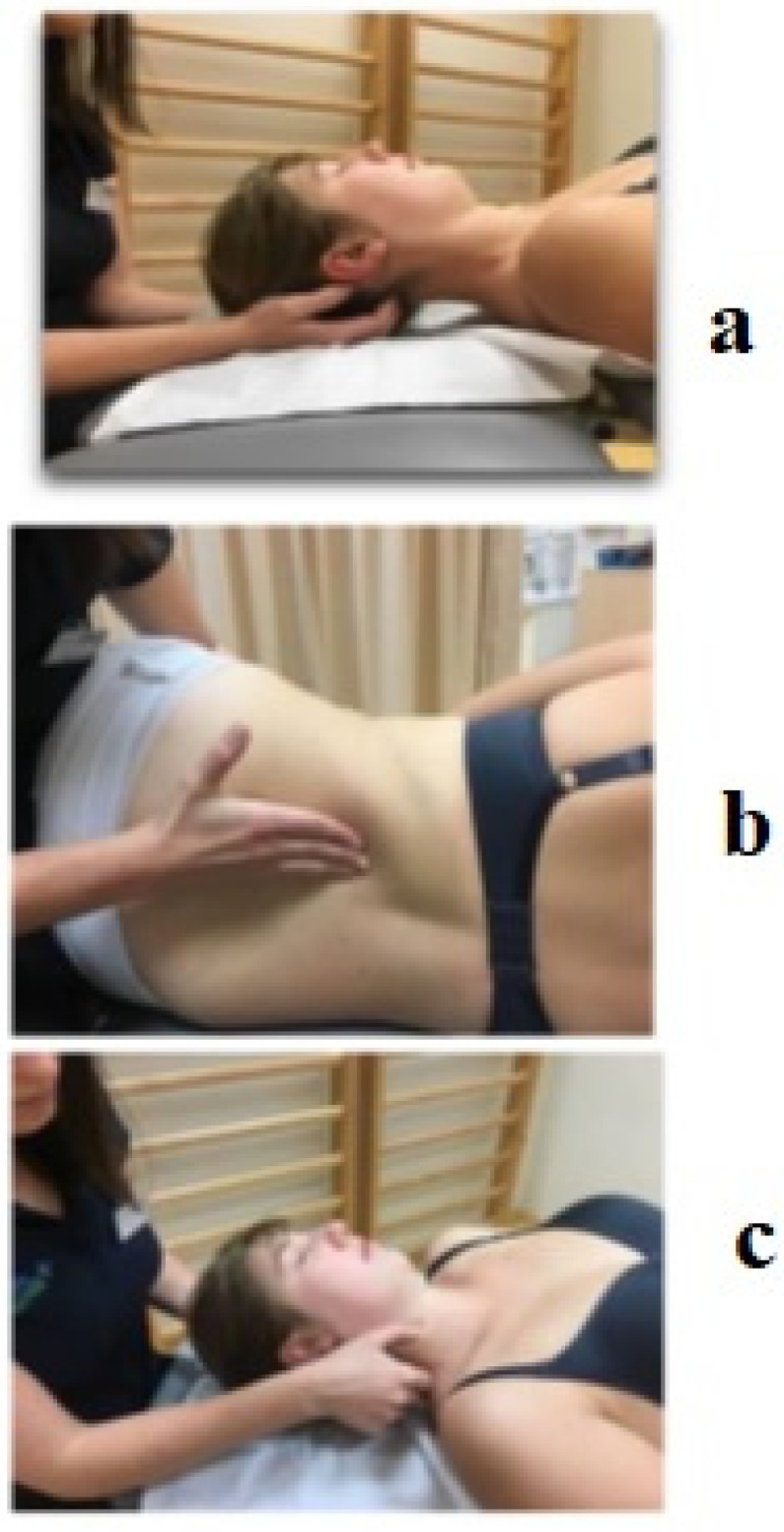
Manual therapy: (**a**)—relaxing the subscapularis muscles, (**b**)—releasing the thoracolumbar fascia, (**c**)—relaxing the sternocleidomastoid muscles (authors’ own source).

**Figure 2 ijerph-20-00369-f002:**
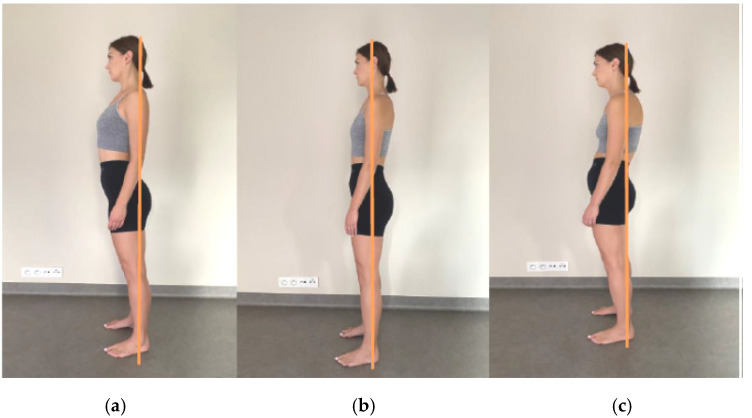
Postural assessment: (**a**)—incorrect posture (excessive forward inclination of the pelvis, hypercorrection of thoracic spine), (**b**)—normal posture, (**c**)—incorrect posture (excessive forward inclination of the spine, thoracic spine in flexion, head protracted) (authors’ own source- model agreement).

**Figure 3 ijerph-20-00369-f003:**
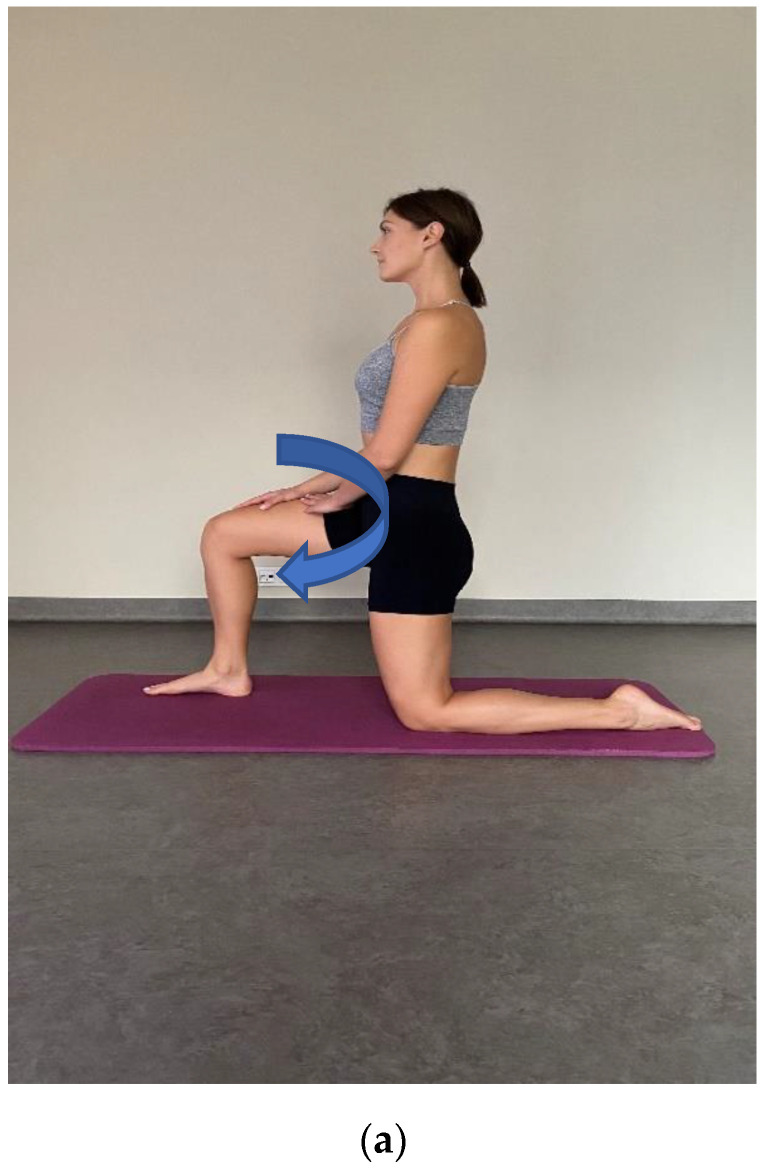
(**a**) Stretching exercise for iliopsoas muscle. (**b**) Stretching exercise for the: latissimus dorsi muscle, quadratus lumborum muscles and thoracolumbar fascia (authors’ own source model agreement).

**Figure 4 ijerph-20-00369-f004:**
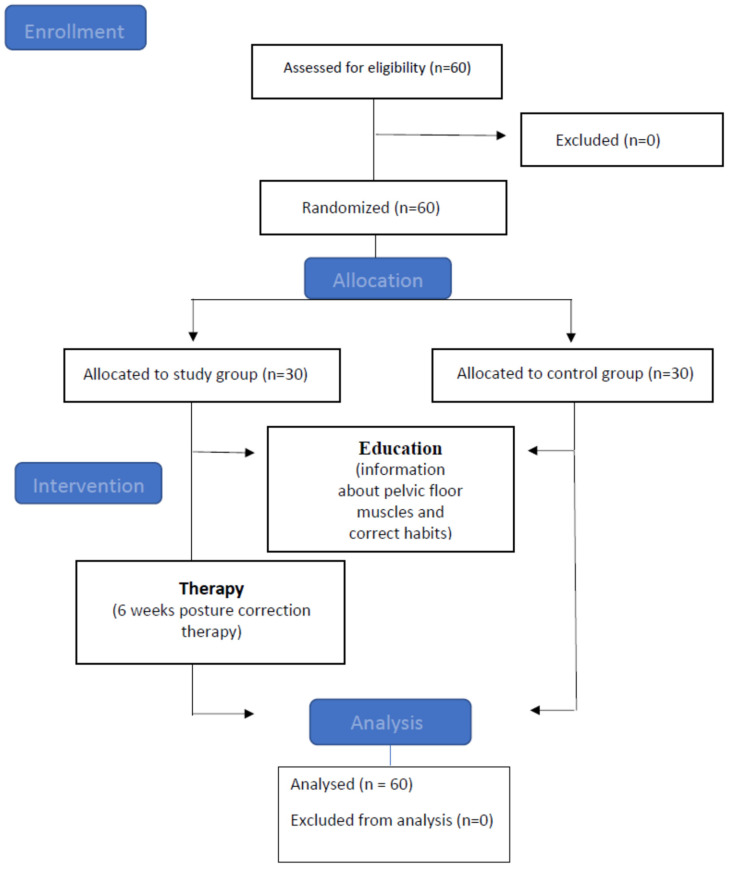
Flow-chart.

**Figure 5 ijerph-20-00369-f005:**
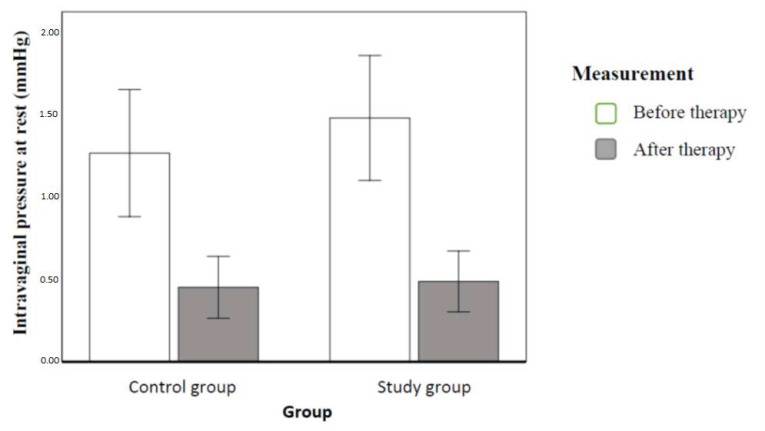
Mean intravaginal pressure with 95% confidence interval in individual groups and order of measurements (authors’ own source).

**Figure 6 ijerph-20-00369-f006:**
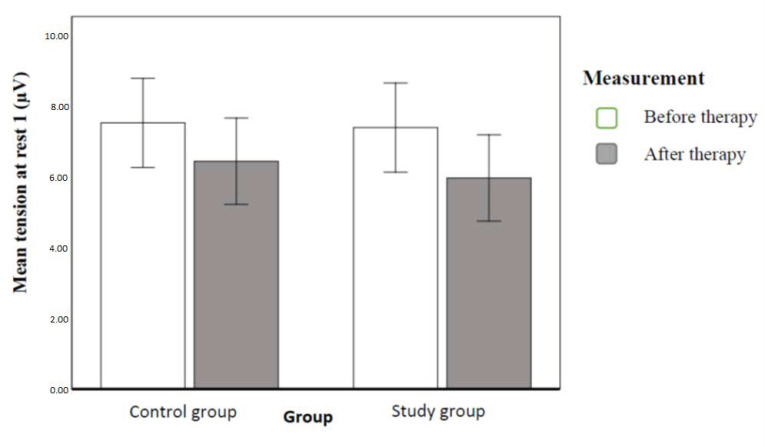
Mean tension at rest with a 95% confidence interval in individual groups and order of measurements (authors’ own source).

**Figure 7 ijerph-20-00369-f007:**
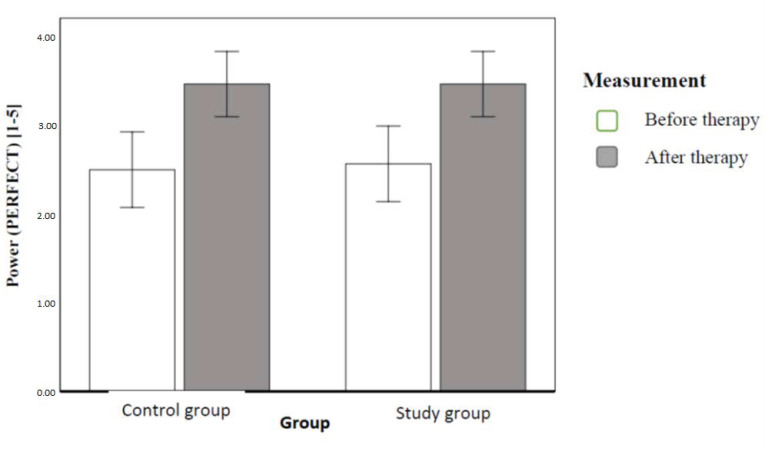
Mean PFM contraction power (Oxford scale) with 95-percent confidence intervals in individual groups and order of measurements (authors’ own source).

**Figure 8 ijerph-20-00369-f008:**
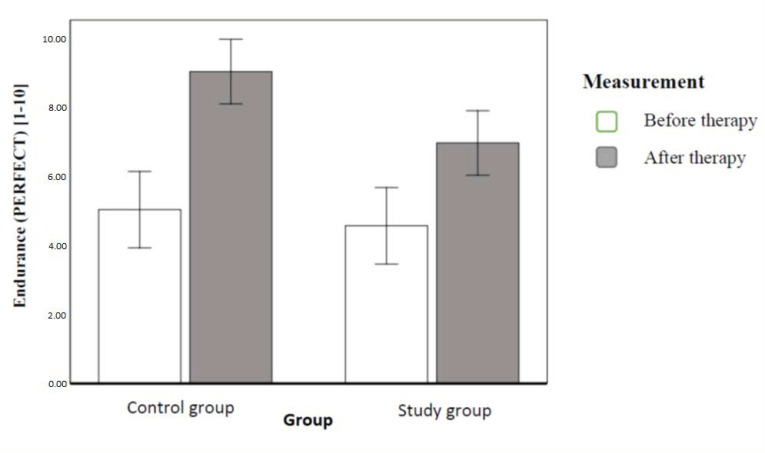
Mean PFM contraction endurance (PERFECT) with 95-percent confidence intervals in individual groups and order of measurements.

**Figure 9 ijerph-20-00369-f009:**
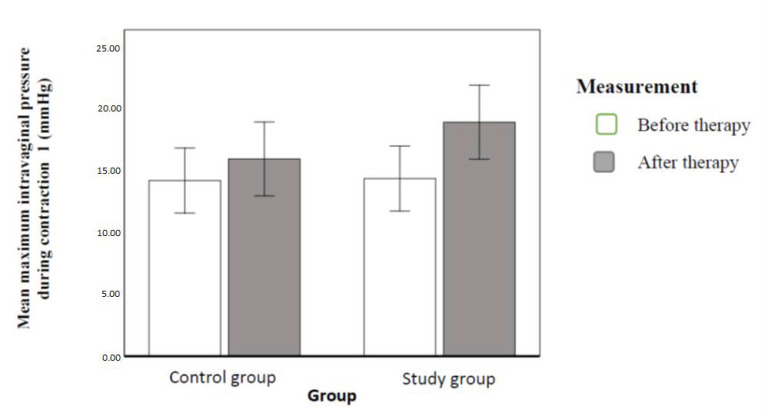
VSP with a 95% confidence interval in individual groups and order of measurements (authors’ own source).

**Figure 10 ijerph-20-00369-f010:**
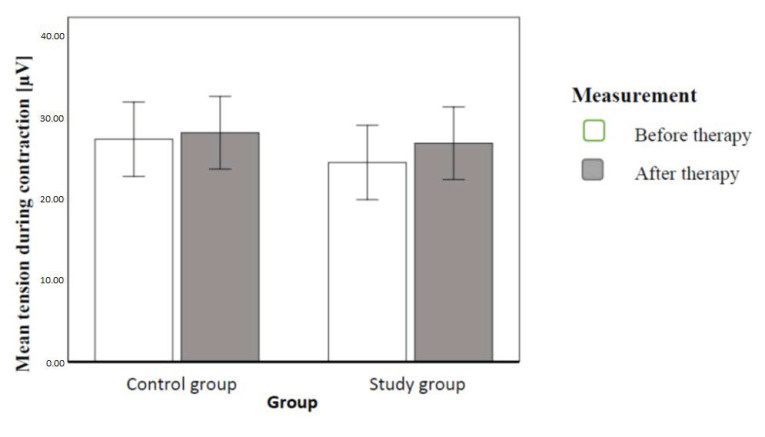
Mean tension during contraction with 95-percent confidence intervals in individual groups and order of measurements (authors’ own source).

**Table 1 ijerph-20-00369-t001:** Descriptive statistics.

Group	Age (Years)	Body Height (m)	Body Mass (kg)	BMI [kg/m^2^]
Study group	38.3 ± 5.54	1.7 ± 0.05	63.4 ± 7.3	22.8 ± 2.6
Control Group	35.5 ± 4.7	1.7 ± 0.04	63.9 ± 0.04	23.2 ± 1.7

**Table 2 ijerph-20-00369-t002:** The PERFECT Scheme (authors’ own design).

Parameter	Description
P (power)	PFM (pelvic floor muscles) power during MVC (maximum voluntary contraction) (the Oxford scale 0–5)
E (endurance)	Time (in seconds) of maintaining MVC (0–10)
R (repetition)	Number of contractions with power P maintaining E for certain amount of time (0–10)
F (fast contractions)	Number of fast contractions, lasting for 1 s, with 1 s long relax between contractions (0–10)
E (elevation)	Elevation of the perineum (yes/no)
C (co-contraction)	Other muscle groups contract during PFM contraction (yet/no)
T (timing)	There is involuntary PFM contraction in the coughing test (yes/no)

**Table 3 ijerph-20-00369-t003:** Essential descriptive statistics along with normality test (authors’ own source).

	M	Me	SD	Sk.	Kurt.	Min.	Maks.	W	*p*
Clinical control group									
Power (PERFECT) 1	2.5	2	1.14	0.68	−0.17	1	5	0.87	0.002
Power (PERFECT) 2	3.47	3	0.94	0.24	−0.73	2	5	0.88	0.004
Endurance (PERFECT) 1	5.03	5	3.22	0.39	−1.13	1	10	0.89	0.005
Endurance (PERFECT) 2	9.03	10	1.67	−1.33	4.73	3	10	0.67	<0.001
Intravaginal pressure at rest 1 (mmHg)	1.32	1.2	1.13	1.3	2.23	0	4.9	0.87	0.002
Intravaginal pressure at rest	0.45	0.5	0.46	1.69	4.81	0	2.1	0.81	<0.001
2 (mmHg)									
Mean tension at rest 1 (µV)	7.52	7.45	3.09	0.2	0.05	1.4	14	0.97	0.456
Mean tension at rest 2 (µV)	6.44	5.6	3.21	1.45	2.85	2	16.7	0.86	0.001
Mean maximum intravaginal pressure during contraction 1 (mmHg)	14.16	14.1	5.87	0.9	0.15	7	27	0.9	0.013
Mean maximum intravaginal pressure during contraction 2 (mmHg)	15.9	15.43	6.2	0.73	−0.1	7.6	29.4	0.94	0.09
Mean tension during contraction 1 (µV)	27.2	24.1	9.66	0.94	0.23	16.1	50.5	0.91	0.022
Mean tension during contraction 2 (µV)	27.99	27.25	8.78	0.71	0.09	17	51	0.95	0.189

M—mean; Mdn—median; SD—standard deviation; Sk.—asymmetry; Kurt.—kurtosis; Min.—minimal value; Maks.—maximal value; D—Shapiro Wilk test result; *p*—statistical significance.

**Table 4 ijerph-20-00369-t004:** Essential descriptive statistics along with normality test.

	M	Me	SD	Sk.	Kurt.	Min.	Maks.	W	*p*
Study population									
Power (PERFECT) 1	2.57	2	1.19	0.68	−0.22	1	5	0.81	0.07
Power (PERFECT) 2	3.47	3	1.07	0.18	−1.19	2	5	0.81	0.077
Endurance (PERFECT) 1	4.57	5	2.82	0.44	−0.62	1	10	0.81	0.069
Endurance (PERFECT) 2	6.97	7.5	3.21	−0.36	−1.62	2	10	0.81	0.072
Intravaginal pressure at rest 1 (mmHg)	1.47	1.45	0.96	0.5	0.32	0	3.9	0.92	0.49
Intravaginal pressure at rest 2 (mmHg)	0.48	0.25	0.54	1.35	1.33	0	2	0.89	0.297
Mean tension at rest 1 (µV)	7.39	7.7	3.78	1.4	4.82	1.2	21.1	0.91	0.418
Mean tension at rest 2 (µV)	5.96	5.25	3.47	1.44	3.07	1.1	17.4	0.97	0.889
Mean maximum intravaginal pressure during contraction 1 (mmHg)	14.32	11.65	8.29	1.67	3.58	4.1	42.6	0.81	0.066
Mean maximum intravaginal pressure during contraction 2 (mmHg)	18.87	16.9	9.74	1.29	1.46	5.4	46.6	0.88	0.247
Mean tension during contraction 1 (µV)	24.36	20.15	14.71	0.94	0.11	8.6	61.6	0.96	0.782
Mean tension during contraction 2 (µV)	26.71	21.85	14.78	0.95	−0.03	9.8	60.5	0.91	0.47

M—mean; Mdn—median; SD—standard deviation; Sk.—asymmetry; Kurt.—kurtosis; Min.—minimal value; Maks.—maximal value; D—Shapiro Wilk test result; *p*—statistical significance.

## Data Availability

Data available on request.

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
