# Peer review of "Posture Correction Therapy and Pelvic Floor Muscle Function Assessed by sEMG with Intravaginal Electrode and Manometry in Female with Urinary Incontinence"

_ijerph, 2022, doi:10.3390/ijerph20010369_

Round 1

Reviewer 1 Report

This study was about posture correction therapy and pelvic floor muscle function assessed by sEMG with intravaginal electrode and manometry in female with urinary incontinence. It is a good study with values. I have a few questions as below.

1.Page 3 line 101 The PERFECT Scheme is not fully reflected in the results of Table 3. It is recommended that the meaning of each component in PERFECT be clearly described at the bottom of Table 2 on page 3.

2.Page 6 The Education section is not included in Figure 3 of the flowchart, and it is recommended that the Education section be added.

3. Results: For the most dependent variables, only within-group differences have been found in the study, and there were no significant differences between groups. The significance should be noted in all the figures. Also, the reason should be discussed in more details in the discussion.

4. Discussion: The advantages and limitations were not mentioned in the discussion.

5. The conclusion  should be more cautious based on the results.

Author Response

Answer on 1st review

Dear Reviewers,

We are very happy that you consider our study interesting. Thank you very much for your analysis of our manuscript. We really appreciate your comments and indication of fragments which should be either made more detailed or enhanced. Your valuable remarks will surely contribute to improving the quality of our manuscript. All other changes or clarification were written below in green.

This study was about posture correction therapy and pelvic floor muscle function assessed by sEMG with intravaginal electrode and manometry in female with urinary incontinence. It is a good study with values. I have a few questions as below.

1.Page 3 line 101 The PERFECT Scheme is not fully reflected in the results of Table 3. It is recommended that the meaning of each component in PERFECT be clearly described at the bottom of Table 2 on page 3.

The perfect scheme was described precisely in table 2

Table 2. The PERFECT Scheme (authors’ own design).

PARAMETER

DESCRIPTION

P (power)

PFM (pelvic floor muscles) power during MVC (maximum voluntary contraction) (the Oxford scale 0-5)

E (endurance)

Time (in seconds) of maintaining MVC (0-10)

R (repetition)

Number of contraction with power P maintaining E for certain amount of time (0-10)

F (fast contractions)

Number of fast contractions, lasting for 1 seconds, with 1 second long relax between contractions (0-10)

E (elevation)

Elevation of the perineum (yes/no)

C (co-contraction)

Other muscle groups contract during PFM contraction (yet/no)

T (timing)

There is involuntary PFM contraction in the coughing test (yes/no)

2.Page 6 The Education section is not included in Figure 3 of the flowchart, and it is recommended that the Education section be added

Thank you. The Education section was added to the flowchart.

  1. Results: For the most dependent variables, only within-group differences have been found in the study, and there were no significant differences between groups. The significance should be noted in all the figures. Also, the reason should be discussed in more details in the discussion.

All the statistical differences were presented in result section in table 2 and 3 in the last column. In our opinion it is satisfactory in results presentation. If not we can add these additional information. 

  1. Discussion: The advantages and limitations were not mentioned in the discussion.

We have rewritten discussion adding Value of the study and study limitation section. Thank toy for the suggestion.

Value of the study. No similar studies are available in the literature, so we are unable to refer to any similar results. Nobody before us conducted manometric assessment of this group of muscles before and after posture correction therapy. However, our results are in line with the impact of pelvic floor muscle training on the manometry results (Antonio et al, 2018).

The therapy implemented in the study can be conducted by any professional therapist, regardless of training courses they have completed. Paying attention to body posture and education opens new opportunities for treatment of inter alia stress urinary incontinence. It gives a chance for making therapy more effective.

Study limitation. For future studies, it seems worth to arrange pelvic floor muscle assessment at the same time of the day. PFM may function differently in the afternoon hours (overload after the whole day) than in the morning hours. It may also be worth to enlarge the studied population, organizing a multi-centre study and to prolong therapy to a minimum of 12 weeks.

  1. The conclusion  should be more cautious based on the results.

Conclusions were rewritten

Reviewer 2 Report

Dear Authors, I had the pleasure of reading your study and was pleasantly surprised by the good quality of the presentation and the clarity of the contents. The only doubt that comes to mind concerns the description of the methods of posture correction: in the Materials and Methods section in particular, there is a good description of the exercises used to correct posture, however the description of the manual therapy methods applied is very general and not very detailed. I believe that, since there are different techniques and schools of thought in the field of manual therapy (more or less supported by the literature), it would be useful to specify a little more how the manual therapy applied in the study was performed, possibly with a brief description techniques or, possibly, by referring to specific working methods described by other authors and integrated into your study.

Apart from this modification, the article seems to be quite good and, in my opinion, it seems valid enough to be published in this journal.

Author Response

Answers on 2nd review

Dear Reviewers,

We are very happy that you consider our study interesting, with high quality. Thank you very much for your analysis of our manuscript.

We have added detailed description of the manual therapy methods applied as well as figure with some soft tissue techniques.

We have that now out manuscript meet standards for publication

Dear Authors, I had the pleasure of reading your study and was pleasantly surprised by the good quality of the presentation and the clarity of the contents. The only doubt that comes to mind concerns the description of the methods of posture correction: in the Materials and Methods section in particular, there is a good description of the exercises used to correct posture, however the description of the manual therapy methods applied is very general and not very detailed. I believe that, since there are different techniques and schools of thought in the field of manual therapy (more or less supported by the literature), it would be useful to specify a little more how the manual therapy applied in the study was performed, possibly with a brief description techniques or, possibly, by referring to specific working methods described by other authors and integrated into your study.

Apart from this modification, the article seems to be quite good and, in my opinion, it seems valid enough to be published in this journal.

Thank you very much

Round 2

Reviewer 1 Report

NA